# Prenatal care providers' perceptions of the SARS-Cov-2 vaccine for themselves and for pregnant women

Philippe Deruelle[1,2]* , Camile Couffignal[3,4] , Jeanne Sibiude[2,4,5], Alexandre J. Vivanti[2,6], Olivia Anselem[2,7], Dominique Luton[2,8,9], Alexandra Benachi[2,6], Laurent Mandelbrot[2,4,5], Christelle Vauloup-Fellous[2,10], Anne Gael Cordier[2,6], Olivier Picone[2,4,5,11]

1 Pôle de gynécologie Obstétrique, Hôpitaux Universitaires de Strasbourg, Strasbourg, France, 2 Groupe de Recherche sur les Infections pendant la Grossesse (GRIG), Vélizy, France, 3 Clinical Research, Biostatistics and Epidemiology Department, AP-HP, Hôpital Bichat, Paris, France, 4 Université de Paris, INSERM, IAME, Paris, France, 5 Assistance Publique-Hôpitaux de Paris, Hôpital Louis Mourier, Service de Gynécologie-Obstétrique, Colombes, France (OP LM), 6 Hôpital Antoine Béclère, AP-HP, Université Paris Saclay, Clamart, France, 7 Maternité Port-Royal, Hôpital Cochin, AP-HP. Centre-Université de Paris, Paris, France (OA), 8 Service de gynécologie-obstétrique, Paris University, FHU Prematurity, Bichat Hospital Assistance publique-Hôpitaux de Paris, Paris, France, 9 INSERM U1016, Institut IMAGINE, Paris, France, 10 Department of Virology, AP-HP, Hôpital Paul-Brousse, University Paris Saclay, Villejuif, France, 11 FHU PREMA, Paris, France

☯ These authors contributed equally to this work.
* philippe.deruelle@chru-strasbourg.fr

## Abstract

### Background

Prenatal care providers will play an important role in the acceptance of SARS-Cov-2 vaccination for pregnant women.

### Objective

To determine the perceptions of French prenatal care providers: midwives, general practitioners (GPs) and obstetricians and gynaecologists (Ob-Gyn) regarding SARS-CoV-2 vaccination during pregnancy.

### Study design

An anonymous online survey was sent to members of French professional societies representing prenatal practitioners. The participants were asked to answer questions on their characteristics and give their opinions of the SARS-CoV-2 vaccine for themselves and women who are pregnant or willing to become pregnant.

### Results

Access to the survey was opened from January 11th, 2021, to March 1st, 2021. A total of 1,416 responses were collected from 749 Ob-Gyn, 598 midwives and 69 GPs. Most respondents (86.7% overall, 90.4% for Ob-GYN, 81.1% for GPs and 80.1% for midwives) agreed

**Data Availability Statement:** The full data are available at: https://drive.google.com/file/d/1-RGy8RuAQDRpOmkRbbVLhCKGgTZhrCBR/view?usp=sharing.

**Funding:** The authors received no specific funding for this work.

**Competing interests:** The authors have declared that no competing interests exist.

to receive the SARS-CoV-2 vaccine. Vaccination against SARS-CoV-2 would be offered to pregnant women by 49.4% 95%CI [48.1–50.8] of the participants. Midwives were less likely to recommend vaccination than GP and Ob-Gyn (37.5%, 50.7% and 58.8%, respectively). The multinomial logistic regression revealed that being an obstetrician, working in a group, usually offering a flu vaccine and wanting to be vaccinated against SARS-CoV-2 were positively associated with considering pregnant women for SARS-CoV-2 vaccination.

## Conclusion

Most French prenatal healthcare providers are favourable towards vaccinating pregnant women, but a large minority express reservation. More evidence on safety and involvement by professional organisations will be important to encourage the access of pregnant women to vaccination against SARS-CoV-2.

## Introduction

Pregnant women with COVID-19 are at an increased risk of adverse pregnancy outcomes and severe forms of COVID-19 [1–4]. Although preventive measures can significantly decrease SARS-CoV-2 transmission, vaccination is the most promising strategy for combatting COVID-19 [5, 6].

Several international scientific societies strongly recommend that pregnant women have access to SARS-CoV-2 vaccines in all phases of future vaccine campaigns [7–9]. Here, vaccination counselling should balance the available data on vaccine safety and individual risks to pregnant women from COVID-19 infection [10]. A worldwide survey interviewed a total of 19,519 adults between July 24, 2020, and August 7, 2020. Globally, 74% agreed that they would get a SARS-CoV-2 vaccine should it become available, while 26% disagreed. In Europe, the rates of acceptance ranged from 85% in the UK to 56% in Poland [11]. These rates might be lower for pregnant women [12, 13] because immunisation is usually not well accepted by obstetricians and pregnant women—despite recommendations—because of concerns about the lack of safety data on the vaccination of pregnant women [14–16].

Research has demonstrated that strategies based on practitioners' involvement are effective in increasing immunisation rates [17]. However, to adapt the content of educational pieces before improving knowledge of provider, it is recommended to conduct a survey on healthcare workers to understand their views. Thus, we aimed to collect answers of a survey from health care workers (obstetricians, midwives and GPs) involved in the management of pregnant women to evaluate their perceptions of the SARS-CoV-2 vaccination.

## Material and methods

A link to an anonymous online survey was sent on January 11th to the members of the French College Obstetrician and Gynaecologist (CNGOF), French College of Foetal Ultrasonography (CFEF), Federation of French Perinatal Health Networks (FFRP), research group of infection and pregnancy (GRIG), Federation of Prenatal Diagnosis Centres (CPDPN), Union of French Obstetricians and Gynaecologists (SYNGOF) and French College of Midwives (CNSF). The link was also published on LinkedIn®, Twitter® and Facebook®. The study began before the release of the "Haute Autorité de Santé" statement that recommended SARS-CoV-2 for high-risk pregnant women [18].

The questionnaire (S1 Fig) included questions on practitioners' demographic information (gender, year of diploma, profession, place and mode of professional exercise) and their flu vaccination habits for themselves and their patients ("yes", "yes only for pregnant women with risk factor i.e. age over 35, obesity, diabetes and essential hypertension", or "no"). The participants were asked if they would be vaccinated themselves and if they would offer the vaccine against SARS-CoV-2—and which type—to pregnant women or those willing to become pregnant and, if not, the reasons.

All statistical analyses were performed by a dedicated statistician using R software (R Foundation for Statistical Computing, Vienna, Austria. http://www.r-project.org/) v. 4.0. For quantitative variables, descriptive statistics used the median and interquartile range. The discrete variables are presented as number and percentages. Missing data were not replaced. For the variables of interest, the confidence interval was estimated to be at 95%. To detect and represent the underlying structures in the data set [19], a multiple correspondence analysis for nominal categorical data was performed on the predefined set, including nine covariates (gender, profession, experience, practice, habits on flu vaccination for themselves or pregnant women, opinions on SARS-CoV-2 vaccination for themselves and pregnant women or women willing to be pregnant). Then, the association between the prescription of SARS-CoV-2 vaccination to a pregnant woman (three modalities of response) and the prenatal caregiver's characteristics and their behaviour to the flu vaccination was assessed by a multinomial logistic regression using a first bivariate analysis (only for the two major groups of Ob-Gyn and midwives). The expected response rate was estimated at 30% among the members of the medical societies (around 10,000 members) for an expected margin of sampling error fixed at 1%. The study protocol was approved by the Institutional Review Board -IRB 00006477- of HUPNVS, Paris 7 University, AP-HP (N° CER-2021-67). When answering the survey, the participants agreed to participate to the study.

## Results

Access to the online survey was opened from January 11th, 2021, to March 1st, 2021. A total of 1,416 participants completed the survey, including 749 obstetricians and gynaecologists (Ob-Gyn), 598 midwives and 69 general practitioners (GPs). Full data are available at https://doi.org/10.17026/dans-25w-r4wf. Table 1 shows the participants' characteristics. The respondents were predominantly women (76.3%) and working for a median of 17 years [8–29] since graduation. All French departments were represented, with 43% of the respondents being from the Paris area. The main modality of exercise was private practice (32.7%), followed by university hospital activity (26.2%) and general hospital activity (17.8%). Most respondents (78.4% 95% CI [77.3–79.5]) had been vaccinated themselves against flu. Midwives were less likely to be vaccinated than Ob-Gyn and general practitioners (68.9, 85.4 and 84.1% respectively). Most of the participants usually prescribe flu vaccine to pregnant women (86.2% 95%CI [85.2–87.1]).

Table 2 presents the respondents' perceptions of the SARS-CoV-2 vaccine. A large majority (86.7% 95%CI [85.7–87.6]) would agree to be vaccinated against SARS-CoV-2. Among those who did not want to receive the vaccine, the main reasons were the lack of data on adverse effects (31.7%) or effectiveness (26.3%), the need for information from professional societies (91, 21.7%) or from other sources (44, 10.5%) or a greater fear of the vaccine than SARS-CoV-2 itself (40, 9.5%). About half of the participants would not offer the SARS-CoV-2 vaccine to pregnant women (50.6% 95%CI [49.2–51.9]). The rate was higher among midwives (62.5% 95%CI [60.6–64.5]) and GP (49.3% 95%CI [43.3–55.3]) compared to Ob-Gyn (41.1% 95%CI [39.3–42.9]). Most of those who answered 'no' were waiting for extra information from medical societies (25.6%), from other sources (14.7%) and from the Ministry of Health (13.3%);

**Table 1. Prenatal caregivers' demographic information and flu vaccination habits for themselves and their patients.**

| | All responders | Ob-Gyn | GP | Midwives |
|---|---|---|---|---|
| | **N = 1416** | **N = 749 (52.9)** | **N = 69 (4.9)** | **N = 598 (42.2)** |
| **Gender** (Na = 10) | | | | |
| Women | 1078 (76.3) | 451 (60.7) | 54 (79.4) | 573 (95.8) |
| Men | 328 (23.2) | 292 (39.3) | 14 (20.6) | 22 (3.7) |
| **Time since graduation (years)** | 17 [8–29] | 19 [9–32] | 15 [7–29] | 16 [8–26] |
| **Medical practice** | | | | |
| Group practice | 953 (67.3) | 549 (73.3) | 30 (43.5) | 374 (62.5) |
| University hospital | 371 (26.2) | 203 (27.1) | 2 (2.9) | 166 (27.8) |
| General hospital | 252 (17.8) | 143 (19.1) | 6 (8.7) | 103 (17.2) |
| Clinics | 112 (7.9) | 97 (13.0) | 1 (1.4) | 14 (2.3) |
| Mixed activity | 141 (10.0) | 93 (12.4) | 10 (14.5) | 38 (6.4) |
| other | 77 (5.4) | 13 (1.7) | 11 (15.9) | 53 (8.9) |
| Private practice | 463 (32.7) | 200 (26.7) | 39 (56.5) | 224 (37.5) |
| **Geographic area of exercise** (Na = 32) | | | | |
| Paris area | 589 (42.6) | 203 (27.9) | 34 (50.0) | 352 (59.9) |
| others | 795 (57.4) | 525 (72.1) | 34 (50.0) | 236 (40.1) |
| **Do you usually get the flu vaccine?** | | | | |
| Yes | 1110 (78.4) | 640 (85.4) | 58 (84.1) | 412 (68.9) |
| No | 306 (21.6) | 109 (14.6) | 11 (15.9) | 186 (31.1) |
| **Do you usually prescribe the flu vaccine to pregnant women?** | | | | |
| Yes | 1220 (86.2) | 675 (90.1) | 56 (81.2) | 489 (81.8) |
| No | 196 (13.8) | 74 (9.9) | 13 (18.8) | 109 (18.2) |

Values are given as N (%) or Median [IQR].

Ob-Gyn = Obstetricians and Gynaecologist.

GP = General Practitioners.

here, 11.5% considered that inputs on side effects insufficient. 43.9%, 95%CI [42.6–45.6] would recommend vaccination to women willing to become pregnant, and 27.5%, 95%CI [26.3–28.9] only would recommend it if there were risk factors. The three professions would first offer an mRNA vaccine to pregnant women (overall = 50.6%; Ob-Gyn 49.9%, GP = 50.7%, midwives = 42.2%).

Fig 1 shows the symmetric map resulting from multiple correspondence analyses of the perceptions of the SARS-CoV-2 vaccine dimensions, here adjusted for the prenatal caregiver's characteristics. A group practice was positively correlated with a positive vaccination practice among caregivers and their patients and with a positive perception of the SARS-CoV-2 vaccine for them and their patients (pregnant or to be pregnant). Being a midwife was positively correlated (angle less than 90 degrees) with unwillingness to offer the vaccine to pregnant women or preconception women and being vaccinated against SARS-CoV-2. A particular cluster was observed considering the comorbidities associated with SARS-CoV-2. Answers to the three questions about the perceptions of the SARS-CoV-2 vaccine were affected by the concept of comorbidities, which was positively correlated with the obstetrician's group (angle less than 90 degrees) and negatively with the midwife's group (angle more than 90 degrees). Individual repartition of the three groups of professions.

The bivariate (Table 3A) and multivariate analyses of the two major groups are detailed in Table 3A and 3B. After adjustment, when we compared the answers 'yes' versus 'no' for the vaccination of pregnant women, the main factors associated with not offering vaccination

**Table 2. Practitioners' perceptions of the SARS-CoV-2 vaccine.**

| | All responders | Ob-Gyn | GP | Midwives |
|---|---|---|---|---|
| | N = 1416 | N = 749 (52.9) | N = 69 (4.9) | N = 598 (42.2) |
| **Would you be willing to be vaccinated against SARS-CoV-2?** | | | | |
| No | 87 (6.1) | 23 (3.1) | 8 (11.6) | 56 (9.4) |
| No because I do not have any risk factor for severe SARS-CoV-2 | 102 (7.2) | 34 (4.5) | 5 (7.2) | 63 (10.5) |
| Yes | 1143 (80.7) | 641 (85.6) | 53 (76.8) | 449 (75.1) |
| Yes because I have at least one risk factor for severe SARS-CoV-2 | 84 (6.0) | 51 (6.8) | 3 (4.3) | 30 (5.0) |
| **Reasons if answer no (multiple choice) (Na = 9)** | | | | |
| There is not sufficient data on the effectiveness | 110 (26.3) | 27 (23.7) | 7 (28.0) | 76 (27.2) |
| There is not sufficient data on the side effects | **133 (31.7)** | **42 (36.8)** | **10 (40.0)** | **81 (29.0)** |
| I am more afraid of the side effects of the vaccine than of the disease | 40 (9.5) | 14 (12.3) | 0 (0) | 26 (9.3) |
| I need other sources of information | 44 (10.5) | 7 (6.1) | 3 (12.0) | 34 (12.2) |
| I need information from professional societies | 91 (21.7) | 24 (21.1) | 5 (20.0) | 62 (22.2) |
| **In the current state of knowledge, would you prescribe the SARS-CoV-2 vaccine to a pregnant woman?** | | | | |
| Yes for all pregnant woman | 405 (28.6) | 270 (36.0) | 20 (29.0) | 115 (19.2) |
| Yes but only for pregnant woman with at least one risk factor* | 295 (20.8) | 171 (22.8) | 15 (21.7) | 109 (18.3) |
| No | 716 (50.6) | 308 (41.1) | 34 (49.3) | 374 (62.5) |
| **Reasons if answer No (Na = 9)** | | | | |
| Because of the type of vaccine currently available | 78 (4.0) | 39 (9.1) | 7 (7.8) | 32 (3.0) |
| There is not sufficient data on the effectiveness | 111 (5.8) | 36 (8.4) | 6 (6.7) | 69 (6.5) |
| There is not sufficient data on the side effects | 222 (11.5) | 88 (20.6) | 15 (16.7) | 119 (11.2) |
| I am afraid about a teratogenic effect of the vaccine | 183 (9.5) | 64 (15.0) | 7 (7.8) | 112 (10.5) |
| I am more afraid of the side effects of the vaccine than of the disease | 32 (1.7) | 15 (3.5) | 1 (1.1) | 16 (1.5) |
| I consider that this is not a population at risk of severe form | 39 (2.0) | 27 (6.3) | 3 (3.3) | 9 (0.8) |
| I am waiting for other sources of information | 283 (14.7) | 104 (24.4) | 11 (12.2) | 167 (15.7) |
| I am waiting information from professional societies | **494 (25.6)** | **209 (49.0)** | **20 (22.2)** | **265 (24.9)** |
| I am awaiting information from ministry of health | 257 (13.3) | 104 (24.4) | 8 (8.9) | 145 (13.6) |
| Not recommended by the French health authority for this population | 212 (11.0) | 80 (18.7) | 12 (13.3) | 120 (11.3) |
| other reason | 21 (1.1) | 11 (2.6) | 0 (0) | 10 (0.9) |
| **Would you prescribe the SARS-CoV-2 vaccine to women willing to become pregnant?** | | | | |
| Yes for all pregnant woman | 622 (43.9) | 361 (48.2) | 24 (34.8) | 237 (39.6) |
| Yes only for pregnant woman with risk factor* | 389 (27.5) | 223 (29.8) | 19 (27.5) | 147 (24.6) |
| No | 405 (28.6) | 165 (22.0) | 26 (37.7) | 214 (35.8) |
| **Among the following vaccines, considering that they are all available, choose the one or those that you would prescribe during pregnancy?** | | | | |
| AstraZeneca and Oxford AZD1222 | 430 (13.4) | 273 (14.1) | 25 (17.0) | 132 (11.9) |
| BioNTech-Pfizer | **879 (27.4)** | **516 (26.6)** | **41 (27.9)** | **322 (29.0)** |
| GSK-Sanofi | 615 (19.2) | 347 (17.9) | 27 (18.4) | 241 (21.7) |
| J and J-Janssen Ad23.COV2.S | 343 (10.7) | 215 (11.1) | 19 (12.9) | 109 (9.8) |
| Moderna mRAN-1273 | 744 (23.2) | 451 (23.3) | 35 (23.8) | 258 (23.2) |
| Novavax | 195 (6.1) | 136 (7.0) | 10 (6.8) | 49 (4.4) |

Values are given as N (%) or Median [IQR]; in bold the first modality of response

* Risk factor included age up to 35, obesity, diabetes, essential blood pression.

Ob-Gyn = Obstetricians and Gynaecologist.

GP = General Practitioners.

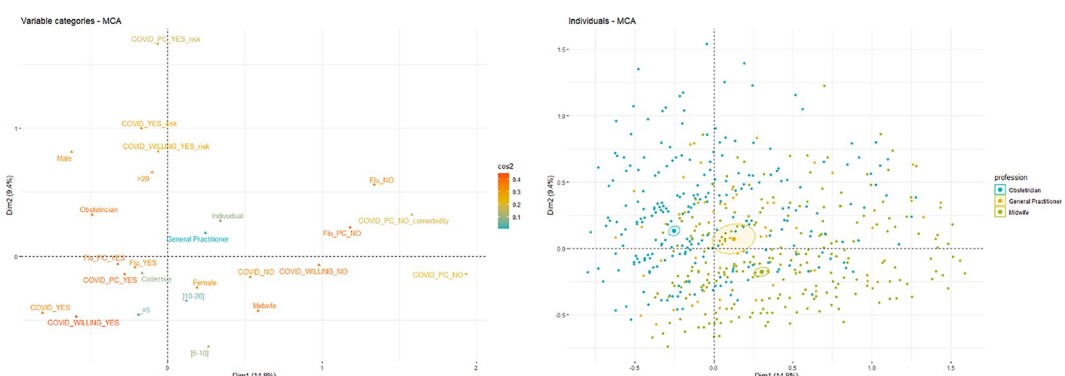

**Fig 1. Multiple correspondence analysis map (projections on the first two dimensions) for prenatal caregiver's answers and the perceptions of the SARS-Cov-2 vaccine for variable variables (left) and individual (right).** Active variables: gender (woman and man); profession (obstetrician, general practitioner, midwife); experience (≤5,] 5–10],] 10–15] or ≥15); modalities of exercise (collective or individual); practitioners' flu vaccine (yes or no); pregnant women flu vaccine (yes or no); practitioners' SARS-CoV-2 vaccination for themselves (no, no because I do not have risk factors, yes or yes because I have risk factors), pregnant women's SARS-CoV-2 vaccination (yes, yes only for pregnant women with risk factors) and women willing to be pregnant SARS-CoV-2 vaccination (yes, yes only for pregnant women with risk factors). The categories defining the horizontal axis, which explain 15% of the variability, are refusing to be vaccinated and refusing to be vaccinated under the argument of lack of comorbidity (towards the right of the map) and at the opposite the agreement to vaccinate pregnant women and women willing to be pregnant (towards the left). The vertical axis accounts for 9% of the variability of the system, being the one that best discriminates between the categories of the variable 'profession', with obstetricians and gynaecologists and midwives having a negative association (angle value close to 180 degrees).

were being a midwife, a woman, working individually, and not wanting to be vaccinated. When we compared 'yes' and 'yes' with comorbidity, gender and prenatal caregivers' perceptions for themselves, gender remained negatively associated with vaccination, while experience and vaccination for oneself were contributing factors.

## Discussion

Our study demonstrates that half of prenatal practitioners would spontaneously prescribe vaccines against SARS-CoV-2 to pregnant women and would recommend an mRNA vaccine. Being an obstetrician in a group practice, being used to prescribe seasonal influenza vaccines or being supportive of the vaccine for oneself improved intentions of vaccine prescription. Therefore, our study gives a snapshot of the nonvaccinators who must be convinced and should be the target of information and educative actions.

There are numerous barriers to and predictors of the vaccination of pregnant women. A review based on 75 articles listed 25 obstacles from the point of view of patients and 24 from the point of view of professionals, along with 18 facilitating elements [20]. As in our study, barriers from knowledge or awareness included concerns about side effects and/or safety to pregnant patients and foetuses, along with confusion regarding institutional recommendations [20]. Current evidence suggests that the role of healthcare providers is a key factor in vaccination decision making [21]. In the face of emerging vaccine hesitancy, healthcare providers remain the most trusted advisers and influencers of vaccination decisions [22]. This result allows us to be optimistic because most caregivers surveyed were motivated to explain and offer the vaccine to their patients.

We found that the respondents who wanted to be vaccinated themselves were more likely to offer vaccines to pregnant women. This is consistent with the finding regarding another preventive recommendation that providers who regularly took a multivitamin were more than twice as likely to recommend multivitamin use to women of childbearing age [23]. Healthcare

**Table 3. a (bivariate analysis) and b (multivariate analysis).** Association between intention to vaccinate the pregnant women and prenatal caregiver's characteristics for the 1,347 responders (obstetrician and gynaecologists and midwives).

| | Reference Yes vs. No | | | Reference Yes vs. Comorbidity | | |
|---|---|---|---|---|---|---|
| **A Bivariate analysis** | | | | | | |
| **Variables** | **Crude OR** | **$_{95\%}$CI** | **pvalue** | **Crude OR** | **$_{95\%}$CI** | **pvalue** |
| Type of profession (ref Midwife vs. Obstetrician) | 0.35 | [0.27–0.46] | **<0.0001** | 0.67 | [0.48–0.92] | **0.015** |
| Gender (ref woman vs; man) | 0.41 | [0.31–0.55] | **<0.0001** | 0.63 | [0.44–0.89] | **0.009** |
| Time since graduation (years) | 1.00 | [0.99–1.01] | 0.709 | 1.01 | [1.00–1.02] | 0.129 |
| Modalities of exercise (ref Individual vs. collective) | 0.55 | [0.41–0.72] | **<0.0001** | 0.70 | [0.49–0.99] | **0.041** |
| Practitioners' flu vaccine for themselves (ref Yes vs. No) | 2.49 | [1.78–3.50] | **<0.0001** | 1.64 | [1.08–2.48] | **0.020** |
| Pregnant women flu vaccination (ref Yes vs. No) | 2.92 | [1.88–4.52] | **<0.0001** | 1.77 | [1.04–3.02] | **0.036** |
| Practitioners' SARS-CoV-2 vaccination for themselves | | | | | | |
| (ref Yes) | | | | | | |
| No | 9.23 | [3.68–23.15] | **<0.0001** | 2.91 | [0.96–8.78] | 0.059 |
| No I do not have risk factors | 24.50 | [5.97–100.6] | **<0.0001** | 20.99 | [4.93–89.28] | **<0.0001** |
| Yes I have risk factors | 1.62 | [0.91–2.86] | 0.098 | 1.97 | [1.03–3.76] | **0.039** |
| **B. Multivariate analysis** | | | | | | |
| **Variables** | **Adjusted OR** | **$_{95\%}$CI** | **pvalue** | **Adjusted OR** | **$_{95\%}$CI** | **pvalue** |
| Type of profession (ref Midwife vs. Obstetrician) | 0.51 | [0.38–0.70] | **<0.0001** | 0.86 | [0.60–1.24] | 0.415 |
| Gender (ref woman vs. man) | 0.60 | [0.43–0.84] | **0.003** | 0.64 | [0.43–0.95] | **0.025** |
| Time since graduation (years) | 1.00 | [0.99–1.01] | 0.645 | 1.01 | [1.00–1.03] | **0.044** |
| Modalities of exercise (ref Individual vs. collective) | 0.66 | [0.48–0.89] | **0.008** | 0.83 | [0.58–1.20] | 0.330 |
| Practitioners' flu vaccine for themselves (ref Yes vs. No) | 1.09 | [0.73–1.62] | 0.671 | 1.01 | [0.63–1.63] | 0.959 |
| Pregnant women influenza vaccination (ref Yes vs. No) | 1.92 | [1.17–3.15] | **0.010** | 1.34 | [0.74–2.43] | 0.324 |
| Practitioners' opinion on SARS-CoV-2 vaccination for themselves (ref Yes) | | | | | | |
| No | 5.52 | [2.14–14.24] | **<0.001** | 2.40 | [0.77–7.45] | 0.130 |
| No I do not have risk factors | 16.10 | [3.86–67.17] | **<0.001** | 19.07 | [4.40–82.57] | **<0.0001** |
| Yes I have risk factors | 1.66 | [0.92–3.01] | 0.093 | 1.91 | [0.99–3.68] | 0.055 |

* pvalue of wald test using a multinomial logistic regression; in bold the significative pvalue considering the threshold of 0.05.

workers with higher levels of confidence in the benefits and safety of vaccines or who were vaccinated will recommend vaccines to their patients [24, 25]. Although we have these data on other vaccines such as influenza, we can extrapolate similar effects with the SARS-CoV-2 vaccine. However, as a pandemic that is placed in the context of public opinion and that is heavily influenced by the media and social networks, it will be interesting to study these factors of the SARS-CoV-2 vaccine, especially in pregnant women.

Any prescription during pregnancy—more specifically vaccination—is a cause for concern because of the hypothetical risk of teratogenicity or complications for both the mother and new-born. Most SARS-CoV-2 vaccines have acceptable safety profiles and have been found to be efficacious against symptomatic SARS-CoV-2 or severe forms in nonpregnant cohorts [26–30]. There are few data on the evaluation of the vaccine in pregnant women. A recent study on 131 women, including 84 pregnant women, found that SARS-CoV-2 mRNA vaccines generated robust humoral immunity in pregnant and lactating women, with immunogenicity and reactogenicity similar to that observed in nonpregnant women [31]. Fears about vaccines in pregnant women generally have included uncertainty about vaccine safety and a lack of data regarding vaccine risks during pregnancy [32]. Surprisingly, in our study, a vaccine using new

technology was not a limiting factor because in our survey, RNA vaccines were the preferred ones for immunising pregnant women.

Many experts consider that vaccination programmes are threatened by growing concerns among the population regarding the safety and usefulness of vaccines [24, 25]. Vaccine hesitancy does not spare caregivers. In nurses, the vaccine hesitancy prevalence rate was 44% and most often concerned seasonal influenza vaccine and hepatitis B vaccine [33]. Even if worried about SARS-CoV-2, the rates of intention to receive a SARS-CoV-2 vaccine might be low in healthcare workers [34]. In an US survey, which was conducted on all employees of a health care system, in December, before the issuance of vaccine emergency use authorisations by the US Food and Drug Administration, only 55% of healthcare employees considered receiving the vaccine, but as of February 18, 2021, 67.2% have received at least one COVID-19 vaccine dose. In our study, more than 80% of the respondents would consider vaccination for themselves. New vaccines are usually singled out because of a perceived lack of testing for vaccine safety and efficacy [35]. Antivaccine positions are also ideological. A study among Finnish healthcare workers showed that although the majority of healthcare workers had high confidence in vaccination, a notable share reported low vaccination confidence [36]. They questioned the benefits and safety of vaccines, and even expressed distrust in the professional competences and intentions of health professional [36]; however, none of these elements are supported by scientific data. For SARS-CoV-2, the efficacy, duration of protection and side effects are important factors for vaccine acceptance.

In our study, midwives would be less likely to support pregnant women becoming vaccinated against SARS-CoV-2 than obstetricians or GPs. These differences in attitudes across professions are consistent with the findings of previous studies on pregnancy or childhood vaccination [37, 38]. The reasons are multiple, but a majority of the midwives are waiting for guidelines from their society or the ministry of health. They were also more likely to fear the side or teratogenic effects. Vilca et al. found that the most important vaccination barrier for influenza or pertussis during pregnancy was the concern related to the vaccine's adverse events (25.9%), and more midwives than obstetrician-gynaecologists expressed this concern (30.8% vs. 10%, p = 0.02) [39]. In a recent review on midwives' attitudes, beliefs and concerns about childhood vaccination, the authors stated that most midwives supported vaccination although a spectrum of beliefs and concerns emerged [40]. A minority expressed reservations about the scientific justification for vaccination, which focused on what is not yet known rather than mistrust of current evidence. They also suggested that the midwifery model of care was shown to focus on providing individualised care, with patient choice being placed at a premium [40]. In France, midwives were only recently allowed to prescribe and administer a large set of vaccines [41], this may influence their perceptions. Indeed, it has been showed that health care workers who have the right to administer vaccines and who reported that they either discussed or administered vaccines frequently considered vaccines to be more beneficial and safe [36].

The strengths of our study include a large number of respondents nationwide. Our survey was proposed when the first vaccines were available and before the Astra-Zeneca® controversy about an increased risk of thrombo-embolic events, so this could not have affected the results. We cannot exclude that our study has some limitations. Our survey was limited to one country with its own habits and organisation. Nonetheless, SARS-CoV-2 vaccine hesitancy and concerns are global issues. The response rate of the GPs was weak compared with their representation over all French caregivers. However, GPs follow less than 20% of all pregnant women and are usually only seen in early pregnancy [42]. Given our recruitment approach, we were unable to determine the number and characteristics of providers who received the invitation to participate but declined. Likewise, we did not have data on the representativeness of the respondents on all practitioners. Our study reflects acceptability at a given point in time, but

this is likely to vary significantly in the context of SARS-CoV-2 based on recommendations and available data, which regularly give rise to controversy.

## Conclusion

In conclusion, understanding caregivers' perceptions of vaccination against SARS-CoV-2 for pregnant women is important for targeting training. Our study demonstrated that French prenatal healthcare practitioners are convinced of vaccinating pregnant women, but some express reservations that must be overcome. Statements from professional organisations and governmental institutions will be important to encourage offering the vaccine to pregnant women. Improved evidence-based knowledge would reduce fears related to adverse effects. The use of forthcoming publications from countries where vaccinations are more advanced and have reached more pregnant women will be very useful [43].

## Supporting information

**S1 Fig. The S1 Fig details the different questions of the questionnaire.**
(DOCX)

## Acknowledgments

We thank Joelle Bellaisch-Allart, Israël Nisand, Philippe Boukobza, Georges Haddad, Bertrand De Rochambeau, Elisabeth Paganelli, Bernard Bailleux, Jean-Louis Simenel, Béatrice Le Mir and Adrien Gantois, representants of the societies that have agreed to distribute the survey.

We thank Scribendi® for their editorial assistance.

## Author Contributions

**Conceptualization:** Philippe Deruelle, Camile Couffignal, Alexandra Benachi, Olivier Picone.

**Data curation:** Jeanne Sibiude, Alexandre J. Vivanti, Olivia Anselem, Anne Gael Cordier.

**Formal analysis:** Camile Couffignal.

**Investigation:** Philippe Deruelle.

**Methodology:** Philippe Deruelle, Camile Couffignal, Dominique Luton, Christelle Vauloup-Fellous, Olivier Picone.

**Project administration:** Philippe Deruelle, Olivier Picone.

**Supervision:** Alexandra Benachi, Laurent Mandelbrot.

**Validation:** Philippe Deruelle, Christelle Vauloup-Fellous, Olivier Picone.

**Writing – original draft:** Philippe Deruelle.

**Writing – review & editing:** Jeanne Sibiude, Alexandre J. Vivanti, Olivia Anselem, Dominique Luton, Alexandra Benachi, Laurent Mandelbrot, Christelle Vauloup-Fellous, Anne Gael Cordier, Olivier Picone.

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
