## [Decision Letter · Decision Letter 0]

22 Jun 2021

PONE-D-21-15049

Prenatal care providers’ perceptions of the SARS-Cov-2 vaccine for themselves and for pregnant women

PLOS ONE

Dear Dr. Deruelle,

Thank you for submitting your manuscript to PLOS ONE. After careful consideration, we feel that it has merit but does not fully meet PLOS ONE’s publication criteria as it currently stands. Therefore, we invite you to submit a revised version of the manuscript that addresses the points raised during the review process.

We look forward to receiving your revised manuscript.

Kind regards,

Hannah Dahlen, RN, RM, BN (Hons), MCommN, PhD FACM

Academic Editor

PLOS ONE

3. Please state whether you validated the questionnaire prior to testing on study participants. Please provide details regarding the validation group within the methods section.

Additional Editor Comments (if provided):

Reviewers' comments:

Reviewer's Responses to Questions

**Comments to the Author**

1. Is the manuscript technically sound, and do the data support the conclusions?

Reviewer #1: Yes

Reviewer #2: Yes

2. Has the statistical analysis been performed appropriately and rigorously? 

Reviewer #1: Yes

Reviewer #2: I Don't Know

3. Have the authors made all data underlying the findings in their manuscript fully available?

Reviewer #1: Yes

Reviewer #2: Yes

4. Is the manuscript presented in an intelligible fashion and written in standard English?

Reviewer #1: Yes

Reviewer #2: Yes

5. Review Comments to the Author

Reviewer #1: Thank you for the opportunity to review this paper which describes the perceptions of health care providers in relation to COVID-19 vaccination for pregnant women.

The study is very timely given the status of the COVID-19 pandemic.

Overall I think the study is soundly conducted and worthy of publication.

The sample is quite large – 1400 responses including obstetricians, midwives and GPs.

Some terminology requires attention to be clearer. I did not know what ‘collective exercise’ in terms of the medical practice was (this is in Table 1 but also in text). Perhaps this did not translate well. I suggest changing Medical practice to Clinical practice and Collective exercise perhaps to Model of care.

Reviewer #2: Thank you for the opportunity to review this paper that reports a cross sectional survey exploring the perceptions of French antenatal care providers, regarding the COVID-19 vaccine. The paper is generally well written, the authors are congratulated on this contribution to contemporary and much needed evidence. The following feedback is offered with the view to strengthening the manuscript.

Page 11 “However, to adapt the content of educational pieces before improving knowledge of provider (this is unclear- do you mean to inform education for healthcare professionals?)

… it is recommended to conduct a SURVEY on healthcare workers to understand their views.” Thus, we aimed to INTERVIEW the health care workers (obstetricians, midwives and GPs) involved in the management of pregnant women.

The two data collection methods CAPITALISED contradict each other please amend.

Page 12 Title needs correcting – currently reads as MATERIAL AND METHOS –please include ‘D’ in the word Methods

Page 12 - “Haute Autorité de Santé” statement that allowed SARS-CoV-2 for high-risk pregnant women [18]. – do you mean RECOMMENDED COVID-19 vaccination for high- risk pregnant women?

Page 12 – “risk factor i.e. age up to 35,” – do you mean over 35?

Page 12 – “…obesity, diabetes and essential blood pression” – do you mean blood pressure, or essential hypertension?

Page 13 “ their behaviour to the flu vaccination” – do you mean their usual participation in the flu vaccination?

The use of the phrase/ word (collective) exercise throughout the manuscript is unclear- from the context it appears that this might be replaced with the word practice – modality or model of practice/ model of care. A few examples are below – please revise and clarify

Page 13 “ The main modality of exercise was private practice (32.7%), followed by university hospital activity (26.2%) and general hospital activity”

Table 1 is clear – the title ‘collective exercise’ however is not clear - for your consideration- may be better understood as ‘practice setting’, or, is it better as collaborative practice? Suggest English language editorial support

Page 17 “A collective exercise was positively correlated with a positive vaccination

practice among caregivers and their patients and with a positive perception of the SARS-CoV-“

page 18 “ modalities of exercise (collective or individual);” in this instance does collective actually mean collaborative? Does exercise mean practice?

Page 21 “ Being an obstetrician in a collective exercise” – this collective exercise does not make sense

“However, in the context of globalisation and competition between countries, political views can influence vaccine acceptance [32].” – it is not clear what direct relevance this statement has to your study findings – were political views or agents discovered in your findings? If there was no direct link or relevance to your findings, I would suggest removing, otherwise please expand and clarify.

Page 23 “In this survey, which was conducted on all employees of a health care system, in December, before…” – is this referring to your study or another one? If it’s about the study in the previous line then maybe clarify … in this US survey.

Thank you again for your work to produce this great study. I wish you all the best.

6. PLOS authors have the option to publish the peer review history of their article (what does this mean?). If published, this will include your full peer review and any attached files.

Reviewer #1: No

Reviewer #2: No

---

## [Author Response · Author response to Decision Letter 0]

30 Jun 2021

Response to the editors: 

A: We check that the manuscript meets PLOS ONE’s style requirements

A: when answering the questionnaire, the practitioner agreed to participate. The ethical committee agreed to the study. 

3. Please state whether you validated the questionnaire prior to testing on study participants. Please provide details regarding the validation group within the methods section.

A: we did not validate the questionnaire which was constructed by the team. 

A: The full data are available at:

https://drive.google.com/file/d/1-RGy8RuAQDRpOmkRbbVLhCKGgTZhrCBR/view?usp=sharing

Response to the reviewers: 

Reviewer #1: Thank you for the opportunity to review this paper which describes the perceptions of health care providers in relation to COVID-19 vaccination for pregnant women.The study is very timely given the status of the COVID-19 pandemic. Overall I think the study is soundly conducted and worthy of publication. The sample is quite large – 1400 responses including obstetricians, midwives and GPs.

Some terminology requires attention to be clearer. I did not know what ‘collective exercise’ in terms of the medical practice was (this is in Table 1 but also in text). Perhaps this did not translate well. I suggest changing Medical practice to Clinical practice and Collective exercise perhaps to Model of care.

A : Indeed, we did not use the correct terminology. We ask an English-speaking medical doctor from UK who suggest to use the term “group practice. 

Reviewer #2: Thank you for the opportunity to review this paper that reports a cross sectional survey exploring the perceptions of French antenatal care providers, regarding the COVID-19 vaccine. The paper is generally well written, the authors are congratulated on this contribution to contemporary and much needed evidence. The following feedback is offered with the view to strengthening the manuscript. 

Page 11 “However, to adapt the content of educational pieces before improving knowledge of provider (this is unclear- do you mean to inform education for healthcare professionals?)

… it is recommended to conduct a SURVEY on healthcare workers to understand their views.” Thus, we aimed to INTERVIEW the health care workers (obstetricians, midwives and GPs) involved in the management of pregnant women.

The two data collection methods CAPITALISED contradict each other please amend.

A : we change for “we collect answers of a survey from healthcare workers”

Page 12 Title needs correcting – currently reads as MATERIAL AND METHOS –please include ‘D’ in the word Methods

A: change made

Page 12 - “Haute Autorité de Santé” statement that allowed SARS-CoV-2 for high-risk pregnant women [18]. – do you mean RECOMMENDED COVID-19 vaccination for high- risk pregnant women?

A : we change as suggested. 

Page 12 – “risk factor i.e. age up to 35,” – do you mean over 35?

A: yes. Change made

 Page 12 – “…obesity, diabetes and essential blood pression” – do you mean blood pressure, or essential hypertension?

Yes. Essential hypertension. Change made. 

Page 13 “ their behaviour to the flu vaccination” – do you mean their usual participation in the flu vaccination?

The use of the phrase/ word (collective) exercise throughout the manuscript is unclear- from the context it appears that this might be replaced with the word practice – modality or model of practice/ model of care. A few examples are below – please revise and clarify

Page 13 “ The main modality of exercise was private practice (32.7%), followed by university hospital activity (26.2%) and general hospital activity”

Table 1 is clear – the title ‘collective exercise’ however is not clear - for your consideration- may be better understood as ‘practice setting’, or, is it better as collaborative practice? Suggest English language editorial support

Page 17 “A collective exercise was positively correlated with a positive vaccination

practice among caregivers and their patients and with a positive perception of the SARS-CoV-“ page 18 “ modalities of exercise (collective or individual);” in this instance does collective actually mean collaborative? Does exercise mean practice? Page 21 “ Being an obstetrician in a collective exercise” – this collective exercise does not make sense

A: Indeed, we did not use the correct terminology. We ask an English-speaking medical doctor from UK who suggest to use the term “group practice. 

 “However, in the context of globalisation and competition between countries, political views can influence vaccine acceptance [32].” – it is not clear what direct relevance this statement has to your study findings – were political views or agents discovered in your findings? If there was no direct link or relevance to your findings, I would suggest removing, otherwise please expand and clarify.

A: We withdrawn this sentence. 

Page 23 “In this survey, which was conducted on all employees of a health care system, in December, before…” – is this referring to your study or another one? If it’s about the study in the previous line then maybe clarify … in this US survey.

A: Indeed, It was not clear. We change for “in an US survey”. 

Thank you again for your work to produce this great study. I wish you all the best.

---

## [Decision Letter · Decision Letter 1]

30 Jul 2021

Prenatal care providers’ perceptions of the SARS-Cov-2 vaccine for themselves and for pregnant women

PONE-D-21-15049R1

Dear Dr. deruelle,

We’re pleased to inform you that your manuscript has been judged scientifically suitable for publication and will be formally accepted for publication once it meets all outstanding technical requirements.

Kind regards,

Andrea Knittel

Academic Editor

PLOS ONE

Additional Editor Comments (optional):

Reviewers' comments:

Reviewer's Responses to Questions

**Comments to the Author**

1. If the authors have adequately addressed your comments raised in a previous round of review and you feel that this manuscript is now acceptable for publication, you may indicate that here to bypass the “Comments to the Author” section, enter your conflict of interest statement in the “Confidential to Editor” section, and submit your "Accept" recommendation.

Reviewer #1: All comments have been addressed

Reviewer #2: All comments have been addressed

2. Is the manuscript technically sound, and do the data support the conclusions?

Reviewer #1: Yes

Reviewer #2: Yes

3. Has the statistical analysis been performed appropriately and rigorously? 

Reviewer #1: Yes

Reviewer #2: Yes

4. Have the authors made all data underlying the findings in their manuscript fully available?

Reviewer #1: Yes

Reviewer #2: Yes

5. Is the manuscript presented in an intelligible fashion and written in standard English?

Reviewer #1: Yes

Reviewer #2: Yes

6. Review Comments to the Author

Reviewer #1: (No Response)

Reviewer #2: The authors have addressed the feedback which has resulted in a strengthened manuscript. Congratulations on this important work.

7. PLOS authors have the option to publish the peer review history of their article (what does this mean?). If published, this will include your full peer review and any attached files.

Reviewer #1: No

Reviewer #2: No

---

## [Editor Report · Acceptance letter]

3 Sep 2021

PONE-D-21-15049R1 

Prenatal care providers’ perceptions of the SARS-Cov-2 vaccine for themselves and for pregnant women

Dear Dr. Deruelle:

I'm pleased to inform you that your manuscript has been deemed suitable for publication in PLOS ONE. Congratulations! Your manuscript is now with our production department. 

Kind regards, 

on behalf of

Dr. Andrea Knittel 

Academic Editor

PLOS ONE